- Pivotal role of the North African Dipole Intensity (NAFDI) on
- alternate Saharan dust export over the North Atlantic and
- the Mediterranean, and relationship with the Saharan Heat
- Low and mid-latitude Rossby waves
- E. Cuevas<sup>1</sup>, A. J. Gómez-Peláez<sup>1</sup>, S. Rodríguez<sup>1</sup>, E. Terradellas<sup>2</sup>, S. Basart<sup>3</sup>, R.
   D. García<sup>1,4</sup>, O. E. García<sup>1</sup>, and S. Alonso-Pérez<sup>5</sup>
- [1] {Izaña Atmospheric Research Centre (AEMET), Santa Cruz de Tenerife, Spain}
- [2] {SDS-WAS Regional Centre (AEMET), Barcelona, Spain}
- [3] {Earth Sciences Department, Barcelona Supercomputing Centre, Barcelona, Spain}
- [4] {Air Liquide S.A., Delegación de Canarias, Candelaria, Santa Cruz de Tenerife, Spain}
- [5] {Universidad Europea de Canarias, La Orotava, Santa Cruz de Tenerife, Spain}

## 15 Abstract

In this study, we revise the index that quantifies the North African Dipole Intensity (NAFDI), 17 and explain its relationship with the Saharan Heat Low (SHL) and mid-latitude Rossby 18 waves. We find outstanding similarities of meteorological patterns associated with the 19 positive NAFDI and the SHL West-phase on the one hand, and with the negative NAFDI and 20 the SHL East-Phase, on the other hand. We introduce the daily NAFDI index and the daily 21 SHL West-East Displacement Index (SHLWEDI). The Pearson correlation coefficient 22 between the daily SHLWEDI 1-day lagged and the daily NAFDI for the period 1980-2013 20 23 June -17 September is fairly high (r = 0.77). The correlation reduces to 0.69 if the 24 SHLWEDI is not lagged. We observe that the SHL West-phase is significantly more frequent 25 than the SHL East-phase, and that the SHL is more intense during its East-phase. We find positive aerosol optical depth (AOD) anomalies in the Western Sahara during positive NAFDI 26 27 / SHL West-phase, and negative AOD anomalies in the central and eastern Sahara during 28 negative NAFDI / SHL East-phase. A significant positive (negative) NE-SW axis AOD

1 anomaly over the Subtropical North Atlantic for positive (negative) NAFDI is found. 2 Remarkable patterns of positive (negative) AOD anomalies over the tropical Atlantic and the 3 Central-Western Mediterranean during negative (positive) NAFDI are observed. The impact 4 of mid-latitude Rossby waves on NAFDI variations depends on both the amplitude and phase 5 of the Rossby wave at 200-300 hPa, which is quantified in this study by the daily Zonal Wind 6 Anomaly at 300 hPa over South Morocco (ZWA300), and the penetration of the Rossby wave 7 into the lower troposphere, quantified by the daily Omega at 500 hPa over Northwest Algeria 8 (O500). The correlation of both ZWA300 and O500 with NAFDI is significant: 0.48 and 0.53, 9 respectively, when we apply 5-day running means to the time series before calculating the 10 correlation coefficients, and increases to 0.66 when a multi-linear regression is performed. 11 The results suggest that ZWA300 drives almost one day in advance the NAFDI, whereas 12 O500 might be ahead respect to NAFDI less than 12 hours. The power spectra of the NAFDI, 13 SHL, ZWA300 and O500 times series in the intermediate time scale range (between 10 and 14 30 days) show 10 especially intense NAFDI spectral peaks, most of them also present in the 15 SHLWEDI spectrum, finding that for many of the NAFDI/SHLWEDI peaks there is 16 associated an O500 and/or ZWA300 peak. Our results indicate that the modes of oscillation of 17 both the NAFDI and the SHL are driven by those mid-latitudes Rossby waves that go deep 18 enough into the lower troposphere imposing their perturbation to the background 19 meteorological fields. A comprehensive top-down conceptual model is introduced to explain 20 the relationships between the NAFDI, the SHL and the mid-latitude Rossby waves and their 21 impact in dust mobilization and transport in Northern Africa.

22

#### 23 **1** Introduction

24 Northern Africa, and specifically the Sahara desert, is the largest and most active dust source 25 in the world (Ginoux et al., 2004, 2012; Huneeus et al., 2011). There are numerous studies 26 dealing with mineral dust transport from the Sahara to the North Atlantic within the Saharan 27 Air Layer (SAL) (e.g., Prospero and Carlson, 1997; Engelstaedter and Washington, 2007; 28 Haywood et al., 2008, Ben-Ami et al., 2009; Adams et al., 2012; Prospero et al., 2014; Ridley 29 et al., 2014) and its impacts on remote regions, i.e. the Caribbean, and the Americas (Perry et al., 1997; Prospero, 1999; Prospero and Lamb, 2003; Chiapello et al., 2005). The impact of 30 31 Saharan dust over the Mediterranean has also been subject of numerous papers in recent years (e.g., Moulin et al., 1998; Kubilay et al., 2003; Pérez et al., 2006; Gerasopoulos et al., 2006; 32

Basart et al., 2009; Jilbert et al., 2010; Salvador et al., 2014; Marconi et al., 2014) addressing 1 2 its impact on air quality (Rodríguez et al., 2001; Escudero et al., 2007a; 2007b; Querol et al., 2009), and ocean fertilization (Guerzoni et al., 1999; Gallisai et al., 2012; Ravelo-Pérez et al., 3 4 2016). However, dust transport over the North Atlantic and Mediterranean has been 5 approached independently in most of the studies, disregarding the fact that the large pressure 6 centres that modulate dust transport to both regions are basically the same. Both the Saharan 7 heat low (SHL), widely analyzed in recent years (e.g. Lavaysse et al., 2009; 2010a; 2010b; 8 2011; 2013; 2015; Chauvin et al.; 2010), and the North African anticyclone (Rodríguez et al., 9 2015) are pressure centres whose variations in intensity, position and extension are key 10 factors for activating dust emissions and transport to the North Atlantic and Mediterranean.

11 In the last years a great effort has been made to explain the most important dust mobilization 12 mesoscale baroclinic processes driven, somehow, by the SHL. Among them, we could 13 highlight the following mechanisms: dry boundary layer convection (Engelstaedter and 14 Washington, 2007; Lavaysse et al., 2010a, and references therein), density currents (Marsham 15 et al., 2008, and references therein; Schepanski et al., 2009; Chaboureau et al., 2016), lowlevel jets (LLJ) (Knippertz, 2008; Schepanski et al., 2009; Fiedler et al., 2013), strong winds 16 17 and high turbulence associated to the Intertropical Convergence Zone (ITCZ) (Flamant et al., 18 2007; Bou Karam et al., 2008; Canut et al., 2010), and African easterly waves (AEWs) (Jones 19 et al., 2003; Knippertz and Todd, 2010). However, although there is general agreement on the 20 fact that changes in intensity and position of SHL play an important role in the atmospheric 21 regional dust recirculation through the activation of the mesoscale baroclinic processes listed 22 above, little is known about the physical processes that determine the variability in the 23 position and intensity of the SHL. It is generally accepted that dust mobilization is mainly 24 controlled by soil characteristics, surface conditions and surface wind speed. Dry soil 25 conditions can be assumed as constant in central hyper-arid Sahara. Therefore, the main factor 26 that modulates the activation of the multiple dust sources present in this region is the low-27 level wind variability.

Chauvin et al. (2010) found a relation between mid-latitude Rossby waves and the longitudinal position of the SHL. However, they did not explain the underlying physical mechanisms behind this relation. Applying Fourier Analysis to the linearized barotropic vorticity equation for a constant-density atmosphere, with a constant zonal flow, it is straightforward to obtain the dispersion relation for the free barotropic Rossby waves (e.g., Holton, 1992). However, the propagation of Rossby waves in the real atmosphere is a much

more complex issue. According to Pedlosky (1987), in the vertically stratified real 1 2 atmosphere, a Rossby wave may be considered as barotropic if it has the same structure 3 throughout the vertical column, except that the amplitude is affected by an exponential factor 4 depending on height. These considerations apply for a static atmosphere or with a constant 5 zonal velocity (Pedlosky, 1987). The use of more realistic models that consider a zonal 6 background flow depending on latitude and height makes the calculation of Rossby waves 7 much more complex since it requires solving numerically an eigenvalue problem of ordinary 8 or partial differential equations with specific boundary conditions. The forced barotropic 9 vorticity equation has been often used in the literature to study the longitudinal propagation of 10 Rossby waves in the upper troposphere (e.g., Charney and Eliassen, 1949; Hoskins and 11 Ambrizzi, 1993; Ambrizzi et al., 1995). Sometimes the Wentzel-Kramers-Brillouin (WKB) 12 method has been used to try to understand, at least qualitatively, the behaviour observed in 13 complex models and in reanalysis (e.g., Hoskins and Karoly, 1981; Karoly and Hoskins, 14 1982; Karoly, 1983; Hoskins and Ambrizzi, 1993; Petoukhov et al., 2013). Hsu and Lin 15 (1992), Hoskins and Ambrizzi (1993), and Ambrizzi et al. (1995) showed that in the Northern 16 Hemisphere there are waveguides for the propagation of Rossby waves in the upper 17 troposphere. The propagation of Rossby waves through the North Atlantic and North African 18 waveguide and the longitudinal position of the SHL are related each other according to 19 Chauvin et al. (2010).

In this study, we start from the results obtained by Rodríguez et al. (2015) regarding the socalled North African Dipole Intensity (NAFDI). The NAFDI is the difference of geopotential
height anomalies averaged over the subtropics and the tropics close to the Atlantic coast.
Rodríguez et al. (2015) analyzed the relationship between the NAFDI and dust export to the
Atlantic in August by using 28-year measurement data of in-situ dust concentrations at Izaña
Observatory and satellite - TOMS and OMI Aerosol Index based - dust retrievals.

The major objectives of the present study are: 1) review the definition of the NAFDI index and assess the results obtained by Rodríguez et al. (2015) using complementary spatial fields of Aerosol Optical Depth (AOD) from MODIS retrievals and MACC (Monitoring Atmospheric Composition and Climate) reanalysis, and extending the analysis to every month of the summertime (June-August); 2) explore the role played by the NAFDI in dust transport to the Mediterranean basin, and its impact on dust source activation over the Sahara; 3) determine and analyse the physical mechanisms that link the NAFDI and the SHL on a daily

basis, and 4) analyse the variations of Rossby waves (amplitude and phase) in the North-East
Atlantic, identifying on a daily basis the physical mechanisms by which these waves
modulate the NAFDI variations and hence, the SHL phases.

In Section 2, the observational and reanalysis datasets used in the study are described. The main results and discussion are tackled in Section 3: review of the NAFDI definition, dust transport and meteorological patterns associated with the NAFDI phases, and physical relationship between the NAFDI, the SHL and mid-latitude Rossby waves. Finally, in Section 4, the conclusions and the schematic conceptual model from hemispheric to meso-scale atmospheric mechanisms driving dust transport over Northern Africa are presented.

10

#### 11 2 Data and methodology

#### 12 2.1 MACC reanalysis

13 The 10-year MACC reanalysis for 2003–2012 (Innes et al., 2013) has been used in this study. 14 The MACC reanalysis data can be downloaded from the European Centre for Medium-Range http://apps.ecmwf.int/datasets/data/macc-15 Weather Forecasts (ECMWF) at reanalysis/levtype=sfc/. In this study, we have used daily averages of AOD at 550 nm 16 computed from the MACC data at 06, 09, 12, 15 and 18 UTC in the period 2003-01-01 to 17 18 2012-12-31.

19 A detailed description of the initial implementation of the aerosol modules for this reanalysis 20 is given in Morcrette et al. (2009) for the modelling part, and in Benedetti et al. (2009) for the 21 assimilation part. The physical parameterizations for the aerosol processes are modelled using 22 the LOA/LMD-Z model (Boucher et al., 2002; Reddy et al., 2005). However, some 23 modifications to the original schemes were introduced over the years (Morcrette et al., 2011). 24 Five types of tropospheric aerosols are considered in the model: sea-salt, mineral dust, 25 organic and black carbon, and sulphate aerosols. The MACC reanalysis was run at T255L60, 26 which is an approximate 78 km  $\times$  78 km horizontal resolution with 60 vertical levels. Dust is 27 treated as a chemically non-reactive component, which is externally mixed like all other 28 aerosols in the MACC model. The data assimilation system used to produce the MACC 29 reanalysis is based on a 2010 release of the ECMWF Integrated Forecasting System (IFS) 30 (Cy36r1). The system includes a 4-dimensional variational analysis (4D-Var) with a 12-hour 31 analysis window for O<sub>3</sub>, CO, NO<sub>2</sub>, SO<sub>2</sub>, HCHO, and aerosols.

- The output of the MACC reanalysis has been validated by the MACC-II VAL sub-project and
- the latest information has been reported by Eskes et al., 2014. The MACC reanalysis has
- already been used in several aerosol studies including mineral dust (e.g. Bellouin et al., 2013;
- Inness et al., 2013; Cesnulyte et al., 2014; Cuevas et al., 2015; Eskes et al., 2015).

### 5 2.2 Satellite (MODIS and MISR) data

The MODerate resolution Imaging Spectrometer (MODIS) onboard the NASA EOS (Earth
Observing System) Terra and Aqua satellites (Salomonson et al., 1989) provides aerosol
properties over both land (Kaufman et al., 1997) and ocean (Tanré et al., 1997) with a near9 daily global coverage.

In this study we have used AOD-500 nm monthly averages for the period 2003-01-01 to 2012-12-31 from the MODIS Collection 6 atmosphere aerosol products available in the 11 12 NASA LADS ftp server (ftp://ladsweb.nascom.nasa.gov/allData/6/). MODIS Collection 6 13 includes a merged product, which uses Deep Blue (DB) retrievals to fill in gaps in the Dark 14 Target/ocean (DT) dataset, with extended coverage to vegetated surfaces, as well as bright 15 land, and improved surface reflectance models, aerosol optical models, and cloud screening, 16 and simplified quality assurance flags (Hsu et al., 2013; Sayer et at., 2013). The new features 17 of Collection 6 are especially important for our study since much of the geographic domain we use in our analysis covers North Africa, a region in which we have to identify both dust 18 19 transport and dust sources over surfaces of high reflectivity on which the AOD retrieval is 20 difficult.

Multi-angle Imaging Spectro Radiometer (MISR) instrument, flying aboard the NASA Earth Observing System's Terra satellite (<u>http://www-misr.jpl.nasa.gov/</u>), gets a global coverage every 9 days with revisit time between 2 and 9 days depending on latitude. MISR can retrieve aerosol properties over bright desert areas due to its unique capability of multi-wavelength observations at forward and backward directions (Kahn el al., 2010). According to Kahn et al. (2010), between 70 and 75% of the MISR AOD retrievals differ less than 0.05, or 20% from the corresponding AERONET ones.

In section 3.3 we have used monthly-averaged values of the MODIS daily merged AOD product (MYD08\_M3\_V6), and MISR monthly AOD at 555nm (MIL3MAE\_V4), for the period 2003-2012, produced with the Giovanni online data system, developed and maintained by the NASA GES DISC (<u>http://giovanni.gsfc.nasa.gov/</u>).

### 1 2.3 ERA-Interim Reanalysis

- ERA-Interim is the latest global atmospheric reanalysis produced by the ECMWF, extending
- from 1979 to present. ERA-Interim is based on a 2006 release of the IFS (Cy31r2). As the
- MACC reanalysis, it is run at a T255L60 resolution. The system includes a 4-dimensional
- variational analysis (4D-Var) with a 12-hour analysis window. A detailed description of
- ERA-Interim Reanalysis is given in Dee et al. (2011).
- In this study we have used daily-averaged fields of temperature, wind, and geopotential height
- at standard levels from 1000 to 500 hPa in the period 2003-01-01 to 2012-12-31.

#### 9 2.4 NCEP/NCAR Reanalysis

The NCEP/NCAR Reanalysis Project is a joint project between the National Centers for
Environmental Prediction (NCEP) and the National Center for Atmospheric Research
(NCAR). A description of the NCEP/NCAR reanalysis can be found in (Kalnay et al., 1996).

13 In this study we have used NCEP/NCAR daily and monthly-averaged data with a spatial resolution of 2.5° latitude x 2.5° longitude, in the period 1980-2013. No fundamental 14 differences have been found in our analysis when using NCEP/NCAR reanalysis or ERA-15 16 Interim Reanalysis. So, we have used the latter, which has a better spatial resolution, in plots 17 and results focused on Northern Africa linked to AOD outputs from MACC and MODIS 18 retrievals. However, for larger geographical scales and for computing correlations and 19 regressions between NAFDI and other atmospheric parameters we have used the user-friendly 20 NCEP/NCAR reanalysis web page interface (http://www.esrl.noaa.gov/psd/products/) for 21 reasons of easiness and efficiency in calculations.

### 22 2.5 Hysplit trajectories

Two sets of daily 48-hour-long air back-trajectories beginning at 12:00 UTC, with a 1-hour time resolution, were computed using the Hybrid Single Particle Lagrangian Integrated Trajectory Model (HYSPLIT) version 4.0 (Stein et al., 2015). The arrival points were set at 30°N 15°W 700 hPa (subtropical North Atlantic) and 37°N, 4°E 700 hPa (Western Mediterranean), for each of the two sets, respectively.

Daily backward trajectories were calculated for the summer months (June, July and August)
 in the period 2003-2012. Wind fields from the NCEP/NCAR reanalysis meteorological data

3 set (Kalnay et al. 1996) were used. The vertical model velocity was used.

4 The percentage of backward trajectories that passed over North Africa was calculated for each 5 month, and for each summer, in the 2003-2012 period, distinguishing between NAFDI 6 positive and negative phases. In order to calculate this percentage, we calculated the fraction 7 of time that each backward trajectory resides in the geographic sector bounded by the parallels 20°N and 36°N, and the meridians 18°W and 50°E, using a similar approach to that 8 9 used by Alonso-Pérez et al. (2007) for calculating the Saharan Index. When that fraction of 10 time is 25% or greater, the associated backward trajectory is flagged as influenced by African 11 air masses, and its starting day is considered as potentially affected by the African CBL.

12

#### 13 3 Results and discussion

#### 14 **3.1** A review and some remarks on the NAFDI definition

The definition of NAFDI introduced by Rodríguez et al. (2015) was made with the aim to analyze dust transport from the Sahara to the North Atlantic Ocean. Thus, the points chosen to calculate the geopotential height derivative anomaly at 700 hPa were selected over Morocco and Mali, almost along the same meridian (around 7°W). Since the scope of the present study covers a much larger region, we have proceeded to revise the definition of the index that quantifies NAFDI.

21 We have computed the correlation map between the geopotential height at 700 hPa over the 22 selected point over Morocco and the geopotential height field at the same level (Figure 1a). 23 This figure shows that the geopotential heights over the two regions (over Morocco and Mali) 24 used to define the NAFDI correlate positively, and therefore, their variations are not 25 completely independent. Moreover, the associated regression plot (Figure 1b) shows a pattern 26 of isolines similar to that of the correlation plot. This isoline pattern indicates that the 27 geostrophic wind anomaly has not only a westward component, but also a significant 28 northerly component. For these reasons we have decided to improve slightly the definition of 29 the NAFDI index by selecting the meridional point at the same latitude of that chosen by 30 Rodríguez et al. (2015) but shifted eastwards symmetrically to the Greenwich meridian, 31 specifically at 5°-7.5°E (North Nigeria) instead of 6°-8°W. Notice that with the new

32

1 definition, the correlation between the 700 hPa geopotential at the selected points is zero 2 (Figure 1a), that is, the geopotential variations are independent. As shown in the correlation maps between the geopotential height at 700 hPa and both the NAFDI index defined by 3 4 Rodríguez et al (2015) (Figure 1c) and the improved NAFDI index for the latter (Figure 1d), 5 there is a higher correlation on the northernmost point and more negative correlations in the 6 tropical belt. Moreover, the geopotential height derivative anomaly that the improved NAFDI 7 index provides is calculated along a line that is perpendicular to the geostrophic wind 8 anomaly and crosses the core of the SHL, where dust is lifted-up. Further information 9 supporting these facts is provided in Sections 3.2 and 3.3. The corresponding regression plot 10 between the improved NAFDI and the geopotential height at 700 hPa for August months is 11 shown in Supplement S1.

The total dust concentrations ( $dust_T$ ) measured at the Izaña Atmospheric Observatory in 12 13 August months and the NAFDI time series from 1987 to 2014 present indeed a Pearson 14 correlation coefficient (r) of 0.67 when using the original NAFDI index definition (Rodríguez 15 et al., 2015), whereas r = 0.72 when using the improved NAFDI index. These results indicate 16 that the outstanding results from Rodríguez et al. (2015) might be even better by using this 17 improved index. The improved NAFDI index will be referred to as the NAFDI From now on, replacing that established by Rodríguez et al. (2015). The monthly NAFDI values for the 18 19 period 1948-2015 are available at http://izana.aemet.es/dataseries/. In this study the NAFDI is 20 used for grouping and averaging spatial distributions of AOD from both MODIS retrievals 21 and MACC reanalysis and some atmospheric parameters related with atmospheric mineral 22 dust exportation from North Africa to the Mediterranean and the subtropical Atlantic. 23 Specifically, we use monthly average values of the NAFDI for June, July and August in 24 different time periods within the longest used period 1980-2013. These summer months are 25 classified, in turn, into three groups: positive NAFDI (> +0.4), negative NAFDI (

2 3

4 5

6

7 8

9

*Φ<sup>i</sup><sub>Mo</sub>* is the NCEP Reanalysis daily mean geopotential height at 700hPa in central Morocco region (30°N, 5°W) for day 'i'.

-  $\Phi_{Ni}^{i}$  is the NCEP Reanalysis daily mean geopotential height at 700hPa in North Nigeria region (12.5°N, 5°E) for day 'i'

-  $< \Phi >$  indicates average of the geopotential for the considered day (i) of the year during the reference period 1981-2010, and subsequent application of a 29-day running mean to the average, in order to remove the small residual random component.

# 3.2 Dust transport and meteorological patterns associated with the NAFDIphases

12 The MODIS AOD anomalies for summer months (June, July and August) with positive and 13 negative NAFDI phases in the period 2003-2012 are shown in Figure 2.

14 Large patterns of AOD anomalies, associated to long-range transport of dust out of Northern 15 Africa, are remarkable. First, we detect a significant positive AOD anomaly that follows a 16 ENE-WSW axis on the subtropical Atlantic under positive NAFDI, which agrees with a 17 positive anomaly of easterly winds between 925 and 700 hPa under positive NAFDI (see 18 Supplement S3), and with results of Rodríguez et al. (2015) using NCEP reanalysis wind data 19 for August in the period 1987-2014. Our results also confirm, and extend its validity to the 20 whole summer period, the relationship between the NAFDI and averaged TOMS and OMI AI 21 data averaged over the so called Subtropical North Atlantic (SNA) found by Rodríguez et al. 22 (2015). This positive (negative) AOD anomaly on the subtropical Atlantic during positive 23 (negative) NAFDI is very small in June, and increases considerably in July and August 24 (Figure 2). A second remarkable pattern is the positive AOD anomaly observed over the 25 tropical Atlantic during negative NAFDI (the opposite under the positive phase). In this case, 26 the anomaly is stronger in June than in July and August. The third major contrast between the AOD patterns in the two phases of NAFDI is found on the Central-Western Mediterranean, 27 28 where positive (negative) AOD anomalies occur during negative (positive) NAFDI. This 29 pattern is especially clear in June, the summer month in which dust intrusions in the Central-30 Western Mediterranean are more frequent (Marconi et al., 2014).