# Peer review of "Pivotal role of the North African Dipole Intensity (NAFDI) on"

_Atmospheric Chemistry and Physics, 2016_

## Author Comment (AC1) · 26 Apr 2016

In Page 1, Line 27:

The sentence "... , and negative AOD anomalies in the central and eastern Sahara during..."

should read: "... , and positive AOD anomalies in the central and eastern Sahara during..."

---

## Referee Comment (RC1) · Anonymous Referee #1 · 9 May 2016

GENERAL COMMENTS

This paper uses re-analysis and satellite data to investigate statistical relationships between aerosol, wind, temperature and geopotential height variations over northern Africa during summer. While the topic itself is interesting as such, I find this work offers rather little to justify publication. The text is extremely long, quite tedious to read and largely confirms things we know already from previous work. The quality of the comprehensive analysis is a little thin in places with no statistical significance of results

provided and some of the index definitions being somewhat arbitrary. The physical interpretations are quite speculative throughout most of the paper. In addition, the structure is not as clear as it should be with many literature results deeply woven into the results section. Therefore, I suggest rejection of this paper. There are ways to bring this to a level sufficient to justify publication, but this would almost mean writing a new paper in my eyes.

MAJOR COMMENTS

1) Length: Everything is unnecessary lengthy about this paper: the title, the abstract, the main text and the conclusions. There are 4 Table and 14 (often multi-panel) figures in the text plus many in the Supplementary Material, which is referred to way too often. In the Introduction you list too many papers for some statements. To get this published, the author should think very hard to concentrate on the essential information and to avoid repetitions.

2) Statistical significance: A lot of statistical results are provided but no significances are given anywhere, not even for simple things such as Pearson correlation coefficients. This needs changing throughout the entire manuscript and only statistical significant results should be discussed in the text.

3) Aim of this work: Until the very end, it has not become entirely clear to me what the purpose of this work really is. I understand that in the end you present a conceptual model of the various elements you have investigated but what is the goal of all this? Is it the dust export you want to understand or something else? Are you trying to build a statistical model that could be used for dust export forecasts even? Are you a attempting a critique of existing work on this region (you do in some places but it is never clearly articulated)? In my eyes, this paper needs a very clear aim and then a clear concept to reach it. At the moment it is quite a mixed bag of things and is therefore hard for the reader to digest.

4) Structure: The Introduction needs to concisely summarise the state-of-the-art.

Based on this, you should explain where you have identified the need for more re-search or a new approach to justify your work. As it stands, a lot of relevant literature results appear late in the paper in the results section, making it hard to read and understand.

5) Method: I find it somewhat bizarre to introduce a new index in a 2015 paper and now read that the same authors change that index again in this work but keeping the name. This will confuse a lot of people, even before your idea could ever be established. In addition, your choice of index seems quite arbitrary to me. Why Morocco and Nigeria as points? Others before you have used EOF or similar to find patterns but your choice is not very well justified in my eyes. Many pressure dipole patterns are mass seesaws with strongly negative correlations (e.g. NAO). The motivation for seeking no correlation between points is not clear to me.

6) Innovation: Throughout the manuscript you keep stating that your results are consistent with Chauvin, Lavaysse, Roehrig, Rodriguez, Varga and others. So what exactly is the progress we are making with this? Plus there is also a sister paper to this by Garcia et al., which I assume may have some overlap. In fact your NAFDI is so highly correlated with the SHL east and west phases that I really wonder whether we need a new index (and thus paper) at all.

7) Physical interpretation: You claim in various places in the paper that you will provide a physical interpretation, where others have not. Most of what you provide then I find quite speculative and hand-waving with rather little in terms of solid evidence. This pertains for example to dust emission, which is nowhere shown. You often infer causality where there is only statistics. In addition, some interesting questions such as the ono why June behaves so differently in many ways than the other two months are not really discussed. Also some of the expressions you use for the physical discussions are a little unclear to me, such as "mesoscale baroclinic processes". What do you mean here?

8) Comparison of parameters: I think you need to reflect more on how independent the various parameters you show are. For example, when comparing NAFDI composites of MODIS and MACC, you seemed to be surprised that they agree so well, but of course these three are directly related to each other with MODIS being assimilated in MACC and MACC geopotential being used to define NAFDI etc. The geopotential is closely related to wind and temperature etc. It is good that you test consistence between ECMWF and NCEP but then you should stick to one and not use them exchangeably.

MINOR COMMENTS

As I think that this paper needs a thorough re-write, I won't give any more detailed minor comments here. Overall, the paper is well written with acceptable English except of a few odd expressions or minor grammatical errors.

---

## Author Comment (AC2) · 13 May 2016

We thank Anonymous Referee # 1 for his/her quick and brief report, although we strongly disagree with his/her suggestion of rejecting the article for publication. Referee#1 does not provide specific scientific arguments for it, but general and ambiguous statements.

We think that the strong decision of recommending no publication needs solid and specific arguments.

Below we answer all and every one of the drawbacks brought by Referee # 1.

GENERAL COMMENTS (GC)

GC-1. This paper uses re-analysis and satellite data to investigate statistical relationships between aerosol, wind, temperature and geopotential height variations over northern Africa during summer. While the topic itself is interesting as such, I find this work offers rather little to justify publication. The text is extremely long, quite tedious to read and largely confirms things we know already from previous work.

On the contrary, we do think that there are several results (of different types) throughout the paper that are relevant enough to justify the publication of the manuscript.

The most outstanding results are summarized here:

1) We have found that NAFDI not only drives dust variability (in terms of AOD anomaly) in summertime over the subtropical North Atlantic but also over the Mediterranean and over the Sahara.

2) We have found outstanding similarities in the meteorological patterns associated with the positive NAFDI and the SHL West-phase, on the one hand, and with the negative NAFDI and SHL East-Phase, on the other hand, suggesting a close relationship between the NAFDI and the SHL. This relationship has been physically explained, and quantified in the present study.

For the latter, we have introduced and used two new daily indexes accounting for the daily variability of the NAFDI and the phase of the SHL, for the period 1980-2013.

3) We have found that the NAFDI is driven by variations in the lower-troposphere wind field at synoptic scale, which are driven, in turn, by the arrival of mid-latitude free barotropic Rossby

waves that impose their perturbed wind and geopotential fields to the state of the lower troposphere. The impact of mid-latitude Rossby waves on NAFDI and SHL variability has been physically explained and statistically assessed using long-term (1980-2013) daily reanalysis data.

As far as the authors know, these results have not been published before. We kindly ask the Referee tell us whether the results presented here have been published previously (in such case, the Referee should provide the exact reference for each result), and / or whether the Referee has scientific arguments that contradict these results.

We think that presenting only one or two of these three results would be more than enough to justify the publication due to their scientific significance.

In relation to the length of the text, we answer this observation in Major Comments (MC-1)

GC-2. The quality of the comprehensive analysis is a little thin in places with no statistical significance of results provided and some of the index definitions being somewhat arbitrary.

We fully agree with the Referee concerning no statistical significance of the results has been provided.

Concerning the statistical results of the NAFDI-SHL and Rossby Waves-NAFDI daily relationships addressed in Sections 3.3. and 3.4 for the summers of the period 1980-2013, we assumed that the fairly good Pearson correlations obtained using more than 3,000 pair-data implied a high statistical significance to "the naked eye". Given the great length of the paper (addressed by the Referee), we thought in focusing efforts and space to describe the physical mechanisms behind these relationships and document the statistical results. However, we agree with the Referee. A complete assessment of statistical significance is presented below in our reply to Major Comment 2 (Part of this assessment will be presented in a new section of the Supplementary Material and in the main text of the manuscript, even if it means extending a little more the paper).

A detailed description of statistical significance of the results is provided in reply to MC-2.

Concerning the indexes, please see reply to MC-5.

GC-3. The physical interpretations are quite speculative throughout most of the paper.

We present two main physical reasonings:

1) We explain how the NAFDI variations drive the phase changes of the Saharan Heat Low (SHL) in page 18 (lines 1-30), after showing the good statistical relationship between them.

2) We explain how the Rossby waves drive the NAFDI variations (pages 24-27). This explanation is indeed enriched with references and links to results from other authors (Chauvin et al., 2010) who reported the arrival of Rossby waves to Southern-Western Europe.

We do not speculate since these physical reasoning are supported by the results of the reanalysis and based on sound physical principles. In case the Referee considers we are speculating he/she should specify exactly where and why.

GC-4. In addition, the structure is not as clear as it should be with many literature results deeply woven into the results section.

The fact there is a large number of references in the result section is due to three reasons:

1) The topics considered in this manuscript are very broad, belonging mainly to two different areas: aerosols and dynamics. So, there are many needed references that cannot be only allocated in the introduction.

2) Facilitate the transition from one section to another, in which quite different issues are addressed.

3) The fact of using numerous results obtained by other authors as a starting point for our own developments and findings.

GC-5. Therefore, I suggest rejection of this paper. There are ways to bring this to a level sufficient to justify publication, but this would almost mean writing a new paper in my eyes.

We think it is not really a fair decision, in view of the significant results we show in the paper, and the arguments presented in this reply. Many of the comments made by Referee#1 are devoid of the minimum necessary scientific argumentation and are arbitrary.

MAJOR COMMENTS (MC)

MC-1.  1) Length: Everything is unnecessary lengthy about this paper: the title, the abstract, the main text and the conclusions. There are 4 Table and 14 (often multi-panel) figures in the text plus many in the Supplementary Material, which is referred to way too often. In the Introduction you list too many papers for some statements. To get this published, the author should think very hard to concentrate on the essential information and to avoid repetitions.

We admit this is a long paper because it deals with several atmospheric issues that are connected to each other, but that can also be considered as belonging to very different fields of study. This makes necessary to introduce the state of the art and an almost separate discussion for each topic. This paper connects atmospheric processes that had been studied independently in the past, providing significant new results and a unifying physical explanation. We have taken care, particularly, in giving detailed information on the data used and calculations performed, so that the methodology is as transparent as possible, and the results might be reproduced by any researcher. This, of course, means a greater extension of the paper. We admit the last section of the paper provides a rather long summary where there are some repetitions of material presented in the result section. We would condense the last section of the paper if the rest of the referees agree. We will also review the length of the results section eliminating possible duplications.

We are struck by the fact the Referee also think the title and the abstract (not particularly long compared to titles and abstracts of other papers) are too long. We do not understand how the Referee can raise this point.

MC-2.  2) Statistical significance: A lot of statistical results are provided but no significances are given anywhere, not even for simple things such as Pearson correlation coefficients. This needs changing throughout the entire manuscript and only statistical significant results should be discussed in the text.

Firstly, please, see above our reply to the comment GC-2. We agree. A complete assessment of statistical significance is presented here for the main results of the paper. Please, indicate if you require specific additional minor results.

**a)** Concerning the correlation plots, the following sentence will be added in the manuscript (Section 3.1):

"The correlation plots shown in this paper are computed using monthly means (only one month per year, e.g., August) of the period 1980-2013 (i.e., 34 years), except for Fig. S5 (2 monthly means per year). Therefore, the critical value for having a significant Pearson´s correlation coefficient ($R$) with a 95% confidence level is 0.34 (i.e., the correlation is significant if $|R|>0.34$)."

Correlation plots are shown in Figs. 1, 11, S5, S6 and S11. The following sentence will be included in the caption of all these plots: "Correlations greater (in absolute value) than 0.34 are significant with a 95% confidence level", except in Fig. S5, for which the critical value is 0.24. All the correlation plots shown have large regions where the correlation is well above this threshold.

**b)** Concerning the correlations between the two NAFDI index versions and the monthly total dust concentration measured at Izaña Observatory for August months (presented in Section 3.1), the number of elements in the time series is 26, and therefore, the critical value for having a significant Pearson´s correlation coefficient ($R$) with a 95% confidence level is 0.39 (i.e., the correlation is significant if $|R|>0.39$).

The correlation of the dust at Izaña Observatory with the old version of the NAFDI index is 0.67 ($r_1$; this value is significant with a 99.98% confidence level), whereas with the new version of the NAFDI index is 0.72 ($r_2$; this value is significant with a 99.997% confidence level). There is a confidence level of 63% about the fact that $r_2$ is significantly larger than $r_1$ .Some details about this confidence level computation will be presented in the Supplement, including the corresponding references; i.e., Fisher transformation, computation of the difference... However, the decision of providing an improved version of the NAFDI index is not based on the improvement of this correlation.

These three confidence levels will be provided in the main text of the manuscript.

**c)** Concerning the correlations between the daily index time series (3,060 values per time series):

Assuming that there is no time-lag autocorrelation in any of the time series, the critical value for having a significant Pearson´s correlation coefficient ($R$) with a 95% confidence level is 0.036 (i.e., the correlation is significant if $|R|>0.036$).

However, indeed, there is time-lag autocorrelation in the time series. We have used the method exposed below to establish an upper bound for the critical value of the Pearson's correlation coefficient. In the four time series (NAFDI, SHLWEDI, O500 and ZWA300) the autocorrelation decreases when increasing the time-lag. The maximum time-lag (MTL) in which there is still a significant autocorrelation (larger than 0.036) is: 22 days for the NAFDI time series, 23 days for the SHLWEDI time series, 7 days for the O500 time series, and 15 days for the ZWA300 time series. Then, to establish an upper bound to the critical value, we consider a lower bound of the number of independent values (LBNIV) in the time series, computed as the ratio between 3060 and MTL. Note that this is a very conservative estimation of the lower bound. For correlations including the SHLWEDI, LBNIV is 133, and the upper bound to the 95%-confidence-level critical values is 0.17. For correlations including NAFDI but not including SHLWEDI, LBNIV is 139, and the upper bound to the 95%-confidence-level critical values is 0.166. However, for correlations not including the former time series but ZW300, LBNIV is 204, and the upper bound to the 95%-confidence-level critical values is 0.137. These explanations will be included in the Supplementary Material.

The correlation ($r_2$) between the 1-day time-lagged SHLWEDI and the daily NAFDI is 0.770 (this value is significant with a 99.999% confidence level) whereas the non-lagged correlation ($r_1$) is 0.688 (this value is significant with a 99.999% confidence level). There is a confidence level of 92.2% about the fact that $r_2$ is significantly larger than $r_1$. Some details about this confidence level computation will be presented in the Supplement, including the corresponding references; i.e., Fisher transformation, computation of the difference…

These three confidence levels will be provided in the main text of the manuscript.

The paragraph starting in line 26 of page 26 will be rewritten as follows (the new text is highlighted here using bold face letter):

"*Table 4 shows the Pearson correlation coefficient between the daily ZWA300, O500 and NAFDI (also for some time lags as well as 5-day running means **-5drm-**). The results led to the following conclusions: 1) the correlation of both ZWA300 and O500 with NAFDI is significant **(with a confidence level larger than 99.999%)**; 2) the correlation between ZWA300 and O500 is low but not negligible **(with a confidence level larger than 86.5%, and 99.7% in case a 5drm is previously applied)**; these two facts together indicate that ZWA300 and O500 are quasi-independent indexes that take into account different aspects of the Rossby wave in agreement with our previous discussion); 3) it seems that ZWA300 drives almost one day in advance the value of NAFDI (**the correlation with NAFDI lagged 1 day is larger than the correlation***

*without any time lag, with a confidence level larger than 58.7%; the correlation with NAFDI lagged 1 day is larger than the correlation with NAFDI lagged -1 day, with a confidence level larger than 72.9%), whereas O500 might be ahead respect to NAFDI less than 12 hours (the correlation with NAFDI lagged 1 day is significantly larger than the correlation with NAFDI lagged -1 day, with a confidence level larger than 79.1%), which agrees what is shown in Figures 5 and 7 of Chauvin et al. (2010): a Rossby wave-packet comes from the Northwest Atlantic and approaches Northern Africa days before a maximum in the SHL displacement is achieved, reaching the centre of the wave-packet Northern Africa when that maximum is achieved ; 4) when applying 5-day running means to the time series before computing the correlation coefficients, they increase significantly (because of the removal of part of the "noise" due to synoptic signal). When performing a multilinear least-square regression of daily NAFDI as function of ZWA300 and O500, a linear correlation of 0.533 is obtained (0.656 for 5-day running means; in both cases, the correlation is significant with a confidence level larger than 99.999%). Supplement S13 provides more details about these regressions.*"

We have noticed there is a minor typo error in Table 4 that will be corrected in the revised version of the manuscript: where it says "NAFDI lagged **1** day          0.309", it should say "NAFDI lagged **-1** day          0.309" (i.e., a minus sign is missing in the lag). Additionally, the correlation between the NAFDI lagged -1 day and ZWA300 will be added to this table: 0.321.

MC-3. 3) Aim of this work: Until the very end, it has not become entirely clear to me what the purpose of this work really is. I understand that in the end you present a conceptual model of the various elements you have investigated but what is the goal of all this? Is it the dust export you want to understand or something else? Are you trying to build a statistical model that could be used for dust export forecasts even? Are you a attempting a critique of existing work on this region (you do in some places but it is never clearly articulated)? In my eyes, this paper needs a very clear aim and then a clear concept to reach it. At the moment it is quite a mixed bag of things and is therefore hard for the reader to digest.

The authors make absolutely clear from the beginning, at the end of the introduction, the motivations and the objectives of the paper, which are reproduced bellow:

1) review the definition of the NAFDI index and assess the results obtained by Rodríguez et al. (2015) using complementary spatial fields of Aerosol Optical Depth (AOD) from MODIS retrievals and MACC (Monitoring Atmospheric Composition and Climate) reanalysis, and extending the analysis to every month of the summertime (June-August). *Note: Rodríguez et al. (2015) study dust transport towards the subtropical North Atlantic in August months.*

2) explore the role played by the NAFDI in dust transport to the Mediterranean basin, and its impact on dust source activation over the Sahara;

3) determine and analyse the physical mechanisms that link the NAFDI and the SHL on a daily basis,

and

4) analyse the variations of Rossby waves (amplitude and phase) in the North-East Atlantic, identifying on a daily basis the physical mechanisms by which these waves modulate the NAFDI variations and hence, the SHL phases.

We have followed a very clear and Cartesian work method, addressing all the above mentioned issues. Moreover, the title is quite self-explanatory and describes perfectly the main objectives of the paper.

So, we do not understand this Referee's comment. Anyway, we reply to each specific question:

I understand that in the end you present a conceptual model of the various elements you have investigated but what is the goal of all this?

Yes. That's correct. The goal is to provide a unifying physical explanation.

 Is it the dust export you want to understand or something else?

Yes, the main purpose of the paper is to understand dust export (using NAFDI), and this leads us to the SHL phase variations and the Rossby waves. We provide a comprehensive physical explanation of the link we have found between all these phenomena. This is useful for understanding dust export, but also it might be very useful for the research groups working on the West African monsoon.  See our reply to GC-1 for more details.

Are you trying to build a statistical model that could be used for dust export forecasts even?

No way. We do not suggest this possibility in the paper.

Are you a attempting a critique of existing work on this region (you do in some places but it is never clearly articulated)?

We use the plot of the second EOF computed by Lavaysse et al. (2013), which is a dipole corresponding to the East-West displacement of the SHL, to select the geographical location of the reference points of the SHLWEDI index. We explicitly stated in our paper: "*We have found that the SHLWEDI values associated to the East-phase (negative NAFDI) are usually larger in absolute value (but less frequent) than those SHLWEDI values associated to the West-phase (positive NAFDI). This seems to be contradictory with the findings of Lavaysse et al. (2010a), that is, just the opposite.*"

Because of these apparent contradictory results, we investigated them from line 16 of page 20 till line 11 of page 22. We explain and clearly articulate why we think the Lavaysse et al. (2010a) SHL statistics might be slightly biased. We also discussed some processes and provide some few suggestions. The purpose there was not to provide a final explanation, but to make constructive suggestions that might stimulate new studies to be performed by other research groups (e.g., groups working on the West African Monsoon).

MC-4. 4) Structure: The Introduction needs to concisely summarise the state-of-the-art. Based on this, you should explain where you have identified the need for more research or a new

approach to justify your work. As it stands, a lot of relevant literature results appear late in the paper in the results section, making it hard to read and understand.

See our replies to MC-3 and GC-4

MC-5. 5) Method: I find it somewhat bizarre to introduce a new index in a 2015 paper and now read that the same authors change that index again in this work but keeping the name. This will confuse a lot of people, even before your idea could ever be established. In addition, your choice of index seems quite arbitrary to me. Why Morocco and Nigeria as points? Others before you have used EOF or similar to find patterns but your choice is not very well justified in my eyes. Many pressure dipole patterns are mass seesaws with strongly negative correlations (e.g. NAO). The motivation for seeking no correlation between points is not clear to me.

Some of the Referee's statements need some clarifications:

1) The NAFDI (North African Dipole Intensity) index was first introduced by Rodríguez et al. (2015).

2) In this paper we propose the first improvement of the NAFDI index in terms of atmospheric processes, and evaluate this improvement through correlation and regression analysis. This is explained in detail in Section 3.1. (page 8).  The Referee could kindly comment why he/she disagrees. So, we don't change the index "again".

3) The choice of NAFDI, as Rodriguez et al. (2015) explain, is based on the difference of geopotential height anomalies between Northern Africa and the African tropical belt, which modulates dust transport into the North Atlantic. So, the NAFDI index quantifies the 700-hPa geostrophic wind anomaly directed towards the subtropical Atlantic on the region between Morocco and Mali. The important issue here is the concept, and a secondary aspect is the specific points selected to calculate the differences.

4) The use of different version of the same index is a common practice (and a reality). Let's consider, for example, the NAO mentioned by the Referee. There are NAO indexes calculated as a difference between points or using EOFs . In fact, most of the NAO indexes have the same name and when using them we need to know the details on how they were calculated. The reasons for having different versions of the same index are several and of different nature: adaptation to certain geographical areas, study applications, simplicity, etc…. The important issue is the physical concept behind the index. For example, it can be used the Hurrell NAO station-based index (between Lisbon-Portugal and Stykkisholmur-Reykjavik) or the Hurrell principal component (PC) NAO index (Hurrell, 1995), or the NAO computed by Jones et al. (1997), which use normalized sea level pressure between Gibraltar and Reykjavik. The Climatic Research Unit (University of East Anglia) provides also NAO data using Reykjavik and Ponta Delgada (Azores). The main concept, the difference of atmospheric pressure at sea level between the Icelandic low and the Azores high, is the important issue. Analogous discussions might be presented for other indexes (i.e., AMO, El Niño/La Niña, …).

Therefore, it is in no way surprising that the same index is calculated in different ways (i.e. choosing different points for the calculation), all of them with the same name.

5) Concerning the Referee comment "*The motivation for seeking no correlation between points is not clear to me*" we would like to add some further explanation in addition to the information already available in the manuscript. NAFDI is mainly directed by the variability of the Morocco anomaly, being the variability in the southern point of the dipole much smaller. The original definition of NAFDI used Bamako as southern point. Since the 700-hPa geopotential height at this point correlates positively with that for Morocco (according to Figure 1a the correlation is in the range 0.2 to 0.3 over Bamako, which is significant at 80% confidence level), using this point means "loosing" part of the geostrophic wind anomaly respect to using Niger (which has no correlation). The latter point has the following additional important advantage: the line that goes from Morocco to Niger is perpendicular to the geostrophic wind anomaly directed by the Morocco variability (this geostrophic wind anomaly is provided by the isolines of the regression plot shown in Fig. 1b).

MC-6 6) Innovation: Throughout the manuscript you keep stating that your results are consistent with Chauvin, Lavaysse, Roehrig, Rodriguez, Varga and others. So what exactly is the progress we are making with this? Plus there is also a sister paper to this by Garcia et al., which I assume may have some overlap. In fact your NAFDI is so highly correlated with the SHL east and west phases that I really wonder whether we need a new index (and thus paper) at all.

We are not sure to have understood to the Referee.

The eigenvalue associated to the second EOF (which is a dipole corresponding to the East-West displacement of the SHL) computed by Lavaysse et al. (2013), which is the equivalent to the daily SHLWEDI we have introduced in our paper, has not been used any more in the literature (according to our knowledge), but something called the SHL intensity. The SHLWEDI can be easily computed (we have done it for the period 1980-2013) without needing performing an EOF analysis over a very restricted set of points (those where the SHL uses to be).

We use the SHL phase (and the NAFDI) to explain the dust transport towards the Mediterranean and the Atlantic. This has not been done before in the literature. The SHL had been studied and related previously with the West African monsoon, but not with dust outbreaks into the Mediterranean and the subtropical North Atlantic. The equivalence between NAFDI and SHL phase is one of the most remarkable results of this paper!

Rodriguez et al. (2015), when introducing NAFDI (and the corresponding index to quantify it) and explaining dust transport patterns over the North Atlantic at the 850 and 700 hPa levels using it, did not conduct any link to the SHL. This has been one of the objectives of the present paper which has culminated with the outstanding result that NAFDI modulates changes in SHL position, and both NAFDI and SHL are modulated, in turn, by mid-latitude Rossby waves.

In order to answer to the Referee's question: *"So what exactly is the progress we are making with this",* we can divide the paper in three main parts, and the final part with the conceptual model. The achievements and novel results and findings in each of these parts are the following ones:

Part 1: Review of the definition of the NAFDI index and assessment of the results obtained by Rodríguez et al. (2015) using complementary spatial fields of AOD from MODIS retrievals and MACC reanalysis, and extending the analysis to every month of the summertime (June-August). New findings about the role played by the NAFDI in dust transport to the Mediterranean basin, and its impact on dust source activation over the Sahara. Nothing about this appears in Rodriguez et al. (2015).

Part 2: Determination and analyses of the physical mechanisms that link the NAFDI and the SHL. For that two new indexes accounting for the daily variability of the NAFDI and the SHL phase have been first introduced to quantify this relationship on a daily basis in the period 1980-2013. All these results have not been previously published, and we think they are robust and quite relevant.

Part 3: Analysis of the variations of mid-latitude Rossby waves (amplitude and phase) in the North-East Atlantic, identifying on a daily basis the physical mechanisms by which these waves modulate the NAFDI variations and hence, the SHL phases. The relationship between NAFDI and the mid-latitude Rossby waves has been quantified on a daily basis for the period 1980-2013. All these results have not been previously published, and we think they are robust and quite relevant.

Final part: Introduction of a conceptual model that links and harmonizes the three previous parts.

We kindly ask the Referee to tell us exactly which of these findings he/she does not agree with, and why.

Concerning the Referee's assumption that *"the sister paper to this by Garcia et al. may have some overlap"*, we have to say that it is not correct. As we have clearly stated in the present paper, the complementary short study carried out by García et al. focuses on the Long-term analysis (1950-2013) on relationships between Saharan dust export over the North Atlantic and the main climate indexes. So, that multi-decadal analysis, in which long-term dust records are crossed with climate indexes, is a completely different (but complementary) approach to those addressed in our paper.

Anyway, we wonder what the Referee means with the term overlap. If he/she were a little bit more explicit we could respond more adequately.

MC.7. 7) Physical interpretation: You claim in various places in the paper that you will provide a physical interpretation, where others have not. Most of what you provide then I find quite speculative and hand-waving with rather little in terms of solid evidence. This pertains for example to dust emission, which is nowhere shown. You often infer causality where there is only statistics. In addition, some interesting questions such as the ono why June behaves so differently in many ways than the other two months are not really discussed. Also some of the expressions you use for the physical discussions are a little unclear to me, such as "mesoscale baroclinic processes". What do you mean here?

Yes. We provided physical explanations to the relationships between NAFDI and SHL displacement, and between the mid-latitude Rossby waves, the NAFDI and the SHL phases. The Referee should enumerate and specify exactly what seems to him/her "quite speculative and hand-waving with rather little in terms of solid evidence" and why, in order we can refute his/her arguments. Otherwise, the Referee is simply discrediting the paper without presenting objective and solid arguments.

Concerning dust emission we do not pretend to explain any physical mechanism. We say literally in the paper the following sentences (Page 31; lines 8-12): "*This is consistent with the fact that a large part of the mesoscale baroclinic mechanisms that produce dust mobilization are associated with SHL displacements and intensity changes. Thus, it seems that NAFDI also modulates largely the activation / deactivation of dust sources in the Sahara, and not just the dust transport out of northern Africa at regional-synoptic scale*.

The distinct behavior of June months, compared with those of July and August, is explained in different parts of Section 3.2, showing why the dust transport patterns over the North Atlantic and the Mediterranean are different to those observed in July and August. This is consistent with the fact that the WAHL (West African Heat Low) is located over the Western Sahara desert from 20 June till 17 September (according to Lavaysse et al., 2009; during such period the WAHL is called SHL). So, during the first two thirds of June the SHL is not present. We will include a sentence concerning this fact in the manuscript.

Concerning the Referee's comment "*You often infer causality where there is only statistics*" Please, note that we have used correlations and time-lagged correlations on a daily basis. So, we can attribute causality from the significant increase of the time-lagged correlations respect to non-lagged correlations. This methodology is explained and used by Screen and Simmonds (2014) in their paper (in Nature Climate Change) when they analyzed weather extremes associated with amplified planetary waves.

Concerning the term "mesoscale baroclinic processes" we refer to several mesoscale processes involved in dust mobilization over the Sahara (density currents, low level jet, strong turbulence associated to the Intertropical Convergence Zone, etc.) referenced in the Introduction of the paper (page 3). All these processes are baroclinic, but we will remove this word from the term in order to avoid any confusion with the synoptic baroclinic instability. So this term will read as "mesoscale processes"

MC-8. 8) Comparison of parameters: I think you need to reflect more on how independent the various parameters you show are. For example, when comparing NAFDI composites of MODIS and MACC, you seemed to be surprised that they agree so well, but of course these three are directly related to each other with MODIS being assimilated in MACC and MACC geopotential being used to define NAFDI etc. The geopotential is closely related to wind and temperature etc. It is good that you test consistence between ECMWF and NCEP but then you should stick to one and not use them exchangeably.

There are mistakes in the referee comment. First, we do not use the MACC geopotential to define NAFDI. Second, MACC reanalysis assimilates Aqua MODIS Dark-target data but not MODIS Deep-Blue, being the latter data the one used in the present work. This means that MACC reanalysis only assimilates data over the ocean and over dark surfaces (with enough vegetation), but not over deserts. Additionally MACC cannot assimilate AOD data over the ocean and the Sahel regions when covered with clouds. Thus AOD data assimilation over these hypothetically suitable regions is, in practice, very limited for many days. So, no data assimilation is performed throughout North Africa (except in southern Sahel). However, we notice that the AOD anomalies patterns from MODIS Deep blue and MACC are quite similar each other even over the Sahara. The reflection suggested by the Referee is made in the first paragraph of page 12.

A comprehensive comparison between AOD from MACC reanalysis and AOD from satellite sensors (MISR, MODIS and OMI), was reported by Cuevas et al. (2015). They addressed the MACC data assimilation issue, and showed that there is fairly good agreement between AOD patterns from MACC and those AOD patterns observed from satellites, including the MISR reference, over the Sahara. The full reference is the following:

*Cuevas, E., Camino, C., Benedetti, A., Basart, S., Terradellas, E., Baldasano, J. M., Morcrette, J. J., Marticorena, B., Goloub, P., Mortier, A., Berjón, A., Hernández, Y., Gil-Ojeda, M., and Schulz, M.: The MACC-II 2007–2008 reanalysis: atmospheric dust evaluation and characterization over northern Africa and the Middle East, Atmos. Chem. Phys., 15, 3991-4024, doi:10.5194/acp-15-3991-2015, 2015.*

Concerning the consistence between ECMWF and NCEP pointed by the Referee, we have seen that at synoptic scale, the scale at which we are working in, both ERA-Interim (ECMWF) and NCEP/NCAR reanalysis provide the same patterns. In fact, the Figures 4, 5, 6, 7, 8, 9 of the manuscript and the Figures S7 and S8 (in the Supplement) were plotted by the authors with both reanalysis, showing exactly the same patterns. We decided to use ERA-Interim in these plots because of its higher spatial resolution compared with NCEP/NCAR. . We will introduce a sentence in the paper concerning the good agreement between both reanalysis.

On the other hand, the use of two independent reanalysis has been carried out in other studies. For example, Knippertz and Todd (2010) used simultaneously ERA-Interim and NCEP/NCAR reanalysis. Chauvin et al. (2010) reported that no fundamental differences were found in their analysis of the intra-seasonal variability of the SHL and its link with mid-latitudes, when considering the 1948-2007 period of the NCEP-NCAR Reanalysis or the 1979-2001 period of ERA-40, even using different time periods.

Following the argument of the Referee, we could conclude that all the parameters/variables in the atmosphere are not independent but closely related between them, since they are related through the set of Partial Differential Equations of the Atmospheric Dynamics, and therefore only one variable should be used/plotted, and there is nothing to be explained.

MINOR COMMENTS As I think that this paper needs a thorough re-write, I won't give any more detailed minor comments here. Overall, the paper is well written with acceptable English except of a few odd expressions or minor grammatical errors.

We are happy the Referee considers the paper has acceptable English.

---

## Author Comment (AC3) · 18 May 2016

**Authors minor change to the manuscript Atmos. Chem. Phys. Discuss. 2016-287 (doi:10.5194/acp-2016-287)**

The manuscript shows in Figure 13 the vertical profile of a barotropic Rossby wave in the absence of vertically-sheared zonal flow in order to compare it with the vertical profile of the Rossby wave obtained in the manuscript for the real atmosphere. The former vertical profile was plotted in the manuscript as proportional to $p^{-1/2}$, whereas indeed it should be proportional to $p^{-2/7}$. This vertical profile, as it was plotted in the manuscript, was indeed only valid for Rossby waves propagating in the vertical direction (i.e., no barotropic). This change does not affect the arguments, results or conclusions of the manuscript.

The minor modifications proposed are described in detail below in order to be strictly rigorous in the manuscript (even for this accessory vertical profile). These minor modifications entail: 1) a small change in the text of the main manuscript; 2) a modified plot of the mentioned vertical profile; 3) Supplement: two changes in the text and addition of a curve in a figure.

**Main manuscript:**

Page 26, lines 6-7. Where it says: õin an isothermal atmosphere with a uniform zonal flowö, it should say: õin an isothermal and static atmosphereö.

Page 58, lines 3. The change indicated in the previous paragraph is also applied to Fig. 13 caption.

Page 58, Figure 13. A new version of this figure replacing the previous one is presented here: the curve called barotropic is now slightly different (it is now $p^{-2/7}$ instead of $p^{-1/2}$).

[Figure]

**Supplement:**

Page 11, lines 19-20. Where it says õIn the absence of vertical shear of the background zonal wind, the function $\psi(z)$ is equal to a constantö, it should say õIn the absence of background zonal wind, the function $\psi(z)$ is equal to $\left(p/p_r\right)^{3/14}$ ö.

Page 12, lines 11-14. Where it says õIn case there is no thermal wind (i.e., the zonal velocity is constant in pressure), $\psi(z)$ becomes a constant, whereas for the external Rossby modes obtained by Geisler and Dickinson (1975) when there is thermal wind (see their Figure 10), $\psi(z)$ has a significant maximum in the lower, middle and/or upper troposphere.ö, it should says õIn the case of no zonal velocity, $\psi(z)$ depends much less on $z$ than in the case with zonal thermal wind (for the latter, see Figure 10 of Geisler and Dickinson, 1975, which corresponds to external Rossby modes; for the former, see the curve denoted as õBarotropicö in Figure S12 of this Supplement). In the latter case (thermal wind), $\psi(z)$ has a relatively prominent maximum in the lower and middle troposphere (see also the curve denoted as õNAFDI Driverö in Figure S12)ö.

Page 23, Figure S12. For reference, we have added a new curve to this figure, corresponding to $\psi(z)$ for the case of no zonal velocity (barotropic).

---

## Referee Comment (RC2) · Anonymous Referee #2 · 21 May 2016

General Comments I state at the outset that I am not a meteorologist although I am quite familiar with aspects of African meteorology related to dust events and much of the literature related to dust and meteorological forcing. So my review is as an informed "user" of meteorology in this field of research. That said, there are aspects of this paper which I do not fully understand – but this may be because of my limited background.

This paper builds on an earlier paper (Rodriguez et al., ACP, 2015) where they introduced the concept of the North AFrican Dipole (NAFD). They developed an index

(NAFDI) that was defined by the relationship of high pressure over the Sahara and low pressures over the tropics. They showed that the large interannual variability in long-term (1987 to 2014) dust concentrations measured at Izaña Observatory (on Tenerife, Canary Islands) could be explained by variations the intensity of the NAFD over the period and that dust export could be related to changes in rainfall patterns and wind fields linked to NAFD. The present paper further examines the role of the NAFD by bringing in a broader range of satellite and meteorological products and by extending the time period of the study. As a result they revise the index that quantifies the North African Dipole Intensity (NAFDI) and show that the Saharan Heat Low (SHL) and mid-latitude Rossby waves play a role in the NAFDI.

The paper addresses an important topic - the factors driving the variability of dust transport out of North Africa. There are interesting aspects to this paper especially as they could eventually be linked to climate variability over time. However the paper is difficult to read. It is too long and detailed. I became lost in the many facets of the discussion. This could be due to the fact that I am not a research meteorologist. There are many aspects of the paper that make sense to me but there are others that I do not understand in the context of the topic.

My recommendation is that this paper has the potential to make a significant contribution to the field but it will require substantial revisions before it is suitable for publication.

MAJOR COMMENTS 1. Length: Aside from the readability problems that might be due to my limitations, there is clearly a tendency in this paper to ramble on. An example is the abstract which is far too long and far too detailed. It would discourage many readers before they reached the body of the paper.

2, Objectives: The background and the objective of the study are not clearly stated. Nor or the conclusions. It should start with the statement of the problem (i.e., the role of dust in climate, the need to understand the response of sources to meteorology and the variability thereof). Then a sentence on the old definition of NAFDI and then

address the effort to revise it and why. The section on p 4, line 25 is a more clear statement of what is done. It could be paraphrased in the abstract. The abstract would end with a discussion of results but much simplified from the present discussion which is too detailed and not understandable by the reader without having read the paper.

3. Significance: In general, it is difficult to relate the NAFDI to real-world results. How does the NAFDI approach relate to other efforts in this field? Many papers address specific meteorological systems that seem to drive dust events. How does the NAFDI relate to these other approaches? There is much reference to statistical metrics to show that the new NAFDI improves on the old. But it is not clear if the improved statistical significance is of practical "significance". For example, on page 29, line 14: "As a result, the total dust concentrations measured at the Izaña Atmospheric Observatory in August months (from 1987 to 2014) and the NAFDI time series for that period show a better Pearson correlation coefficient between them when using the improved NAFDI (0.72 instead of the value 0.67 that is obtained when using the original NAFDI definition)." It is not obvious how this improvement is manifested in a larger sense and how this compares with other efforts to characterize dust export. To me, figures such as Fig. 1, 2, 3, 4, 8, 9, etc. are more persuasive than Fig. 11 and 12 for example.

4. Conclusions. This section provides some interesting insights. But there is a lot of discussion in this section that should not be a part of "conclusions". Many of the insights are lost in the very long and convoluted text.

---

## Author Comment (AC4) · 26 May 2016

We thank Referee #2 for his/her report. We are happy for his/her positive opinion about the content of the paper and for his/her constructive critical comments will help us to improve substantially the manuscript.

**General Comments:**

I state at the outset that I am not a meteorologist although I am quite familiar with aspects of African meteorology related to dust events and much of the literature related to dust and meteorological forcing. So my review is as an informed "user" of meteorology in this field of research. That said, there are aspects of this paper which I do not fully understand – but this may be because of my limited background. This paper builds on an earlier paper (Rodriguez et al., ACP, 2015) where they introduced the concept of the North AFrican Dipole (NAFD). They developed an index (NAFDI) that was defined by the relationship of high pressure over the Sahara and low pressures over the tropics. They showed that the large interannual variability in longterm (1987 to 2014) dust concentrations measured at Izaña Observatory (on Tenerife, Canary Islands) could be explained by variations the intensity of the NAFD over the period and that dust export could be related to changes in rainfall patterns and wind fields linked to NAFD. The present paper further examines the role of the NAFD by bringing in a broader range of satellite and meteorological products and by extending the time period of the study. As a result they revise the index that quantifies the North African Dipole Intensity (NAFDI) and show that the Saharan Heat Low (SHL) and mid-latitude Rossby waves play a role in the NAFDI. The paper addresses an important topic - the factors driving the variability of dust transport out of North Africa. There are interesting aspects to this paper especially as they could eventually be linked to climate variability over time. However the paper is difficult to read. It is too long and detailed. I became lost in the many facets of the discussion. This could be due to the fact that I am not a research meteorologist. There are many aspects of the paper that make sense to me but there are others that I do not understand in the context of the topic. My recommendation is that this paper has the potential to make a significant contribution to the field but it will require substantial revisions before it is suitable for publication.

We agree with the referee. We have focused our major efforts to give consistent, rigorous and detailed arguments supporting our findings and it seems clear that we have not optimized the exposure and wording of the objectives and results, reporting them without enough brevity and clarity. We admit the paper is difficult to read, probably because two very different fields

converge in it: dust-aerosols and atmospheric dynamics. However, this fact should not constitute a problem for re-writing the paper in a smoother way we had done. We are making a substantial revision of the way the article has been written, making it shorter and smoothing it significantly (e.g., moving a lot of secondary details of the results to the Supplement).

MAJOR COMMENTS (MC)

MC1. Length: Aside from the readability problems that might be due to my limitations, there is clearly a tendency in this paper to ramble on. An example is the abstract which is far too long and far too detailed. It would discourage many readers before they reached the body of the paper.

Referee #2 agrees with Referee#1 concerning the length of the abstract; and additionally he/she raises the point that the abstract contains too many details. So, we will remove many details and restructure the text to clearly state the major outcomes, reducing it to about half its length.

MC2. Objectives: The background and the objective of the study are not clearly stated. Nor or the conclusions. It should start with the statement of the problem (i.e., the role of dust in climate, the need to understand the response of sources to meteorology and the variability thereof). Then a sentence on the old definition of NAFDI and then address the effort to revise it and why. The section on p 4, line 25 is a more clear statement of what is done. It could be paraphrased in the abstract. The abstract would end with a discussion of results but much simplified from the present discussion which is too detailed and not understandable by the reader without having read the paper.

Thank you for the suggestions. In the new version of the manuscript, the introduction will start with the statement of the problem, and objectives will be clearly stated and motivated in the introduction, as we identify main gaps in the state of art. We agree the objectives presented in page 4, line25 are not well linked with the present state of the art and the existing gaps. Concerning the abstract, please, see Reply to MC1.

3. Significance: In general, it is difficult to relate the NAFDI to real-world results. How does the NAFDI approach relate to other efforts in this field? Many papers address specific meteorological systems that seem to drive dust events. How does the NAFDI relate to these other approaches? There is much reference to statistical metrics to show that the new NAFDI improves on the old. But it is not clear if the improved statistical significance is of practical "significance". For example, on page 29, line 14: "As a result, the total dust concentrations measured at the Izaña Atmospheric Observatory in August months (from 1987 to 2014) and the NAFDI time series for that period show a better Pearson correlation coefficient between them when using the improved NAFDI (0.72 instead of the value 0.67 that is obtained when

using the original NAFDI definition)." It is not obvious how this improvement is manifested in a larger sense and how this compares with other efforts to characterize dust export. To me, figures such as Fig. 1, 2, 3, 4, 8, 9, etc. are more persuasive than Fig. 11 and 12 for example.

Probably we have not adequately explained the improvement on the criteria used to compute the NAFDI index, giving the impression this improvement is of great importance when it is not. Actually, it is a secondary methodological aspect that refers only to the quantification of NAFDI but it does not affect, or alter at all, the concept of NAFDI established by Rodriguez et al. (2015). In fact, most of section 3.1 will be moved to the Supplementary material.

The new outstanding findings concerning NAFDI, not addressed by Rodriguez et al. (2015), are the following:

1) NAFDI shows intra-seasonal variability. So, in the same summer (same year) there can be months with positive and negative phases.

2) NAFDI phases drive dust transport over the Mediterranean too, and not only over the Atlantic.

3) NAFDI phases modulate the position of the SHL, and therefore some mesoscale processes responsible for dust mobilization in the Sahara which are closely linked to the intensity and position of the SHL. We explain the relationship between NAFDI and SHL providing strong physical arguments and support it with statistical analysis.

4) The changes in NAFDI phase, and thus in the SHL phase, are modulated by the Rossby waves (propagating through the North-Atlantic--North-African waveguide) that penetrate deep enough into the lower troposphere.

We will clarify the significance of these results, which explain intra-seasonal variations from months to days in dust transport towards the Atlantic and the Mediterranean, as well as dust mobilization on the Sahara.

Figures 11 and 12 will be moved to the Supplementary material.

MC-4. Conclusions. This section provides some interesting insights. But there is a lot of discussion in this section that should not be a part of "conclusions". Many of the insights are lost in the very long and convoluted text.

We agree. We will remove the long summary and discussion, and the conclusions will be succinctly exposed. The brief summary concerning the conceptual model will be kept.

Summarizing, the substantial revisions (suggested by the Referee and concerning the way the manuscript has been written) we are performing are the following:

1. A new shorter abstract, re-structured and cleared of details, in which main results will be clearly exposed.

2. A new writing of the introduction following the Referee recommendations

3. Most of the content of Section 3.1 will be moved to the Supplementary material.

4. The text, tables and graphs that have to do with methodological details and secondary arguments in sections 3.2, 3.3, and 3.4 will be moved to the Supplementary material or removed.

5. The present "Summary and Conclusions" section will be shortened drastically becoming a "Conclusions" section. Outstanding results will be briefly enumerated with no discussion, and will be complemented with the final conceptual model.

We will upload the new version of the manuscript as an author comment to the Discussion page, within the next few (3-4) weeks (ideally after the Referee #3 comments are posted), to allow further comments of the Referees.

---

## Referee Comment (RC3) · Anonymous Referee #3 · 28 May 2016

The study analyses the role of the NAFTI on the dust transport and the potential relationship with the Saharan Heat Low and the mid latitude circulation. Even if this paper shows some interesting it is absolutely necessary to make substantive changes before acceptation. So I would recommend major revisions.

Major points : 1) the paper is too long and needs to be drastically reduced. I especially suggest to remove a lot of descriptive comments or too speculative conclusions proposed by the authors in section 3.2, 3.3 and 3.4. For instance L11 end of paragraph 2,

p13 (too descriptive), p17 second paragraph (too speculative), p19 first paragraph (too long and not clear), p21 beginning of the 3rd paragraph (too speculative and out of the scope)

2) there is a big confusion associated with daily and monthly correlations and daily or monthly values. The authors should clarify how they perform the calculation.

3) all the results are presented without significance tests.

4) the definition of the NAFTI is justified with the August correlation. Why June/July and September are not taken into account ?

5) September month is not provided in Fig. 2 to Fig. 9. But the authors discussed the link with the SHL that is defined for the entire rainy season over the Sahel from 20 June to 17 September. Why the NAFTI is studied only from June to August ? is there a scientific reason ? I agree that is this is the common definition of summer, but in a science point of view this is not necessary robust due to the seasonal cycle of the West African Monsoon.

6) there is a constant back and forth between ERAI and NCEP reanalysis. I would suggest to use only one set of data to be consistent. These two reanalysis are relatively closed but difference are still present.

7) the role of the African Easterly Waves, one of the most important component of the west african monsoon that can modulate the wind and temperature fields at 700hPa is not mentioned.

8) the use of the geopotential at 1000 hPa should be used with a lot of caution due to the topography of the region. The value is provide by using an linear interpolation technique that can influence the results.

These majors changes need to done before to analyse in more details the minor comments.

---

## Author Comment (AC5) · 17 Jun 2016

We thank Referee #3 for his/her report. We appreciate his/her constructive critical comments and some interesting suggestions that are helping us to improve substantially the way in which the results are presented in the manuscript.

**General Comments:**

The study analyses the role of the NAFTI on the dust transport and the potential relationship with the Saharan Heat Low and the mid latitude circulation. Even if this paper shows some interesting it is absolutely necessary to make substantive changes before acceptation. So I would recommend major revisions.

We are about to complete major revisions in the manuscript drafting, detailed below.

MAJOR COMMENTS (MC)

MC1. The paper is too long and needs to be drastically reduced. I especially suggest to remove a lot of descriptive comments or too speculative conclusions proposed by the authors in section 3.2, 3.3 and 3.4. For instance L11 end of paragraph 2 p13 (too descriptive), p17 second paragraph (too speculative), p19 first paragraph (too long and not clear), p21 beginning of the 3rd paragraph (too speculative and out of the scope).

We agree the paper is too long. We have almost finished a new version of the manuscript shortened substantially: we have moved some detailed analysis and secondary results of sections 3.1, 3.2., 3.3 and 3.4 to the Supplement material, removed some no necessary material and redundancies, shortened considerably the conclusions, and rewriting completely the abstract and the introduction.  We recognize the level of detail and the large extension of the manuscript had blurred the objectives and very relevant results we think the paper contains. The new version will be uploaded next week.

Concerning the specific comments:

L11 end of paragraph 2 p13 (too descriptive)

In the new version of the manuscript, this information has been moved to the Supplement material since this is not core information.

p17 second paragraph (too speculative)

Probably the information of page number and paragraph given by the Referee#3 is not correct, since we do not find here any possible speculative argument.

p19 first paragraph (too long and not clear)

We have re-written this paragraph adding information concerning the statistical significance.

p21 beginning of the 3rd paragraph (too speculative and out of the scope)

We think the discussion presented in this paragraph is not speculative, but based on solid physical arguments. However, the purpose there was not to provide a final explanation, but to present arguments that might stimulate new studies to be performed by other research groups (e.g., groups working on the West African Monsoon). Anyway, since this discussion represents a secondary observation/assessment of the main line of argument in our study, the full page has been moved to Supplement for the sake of brevity.

MC2. There is a big confusion associated with daily and monthly correlations and daily or monthly values. The authors should clarify how they perform the calculation.

We appreciate the Referee comment.

We have eliminated this confusion in the new version of the manuscript. First, the former section 3.1 has been moved to the Supplement. Second, in the new section 3.1 (old 3.2) we work with NAFDI monthly values as Rodriguez et al. (2015), who used only August months, extending the study period to the whole summer and other geographical regions. Third, in the new section 3.2 (old 3.3), when we investigate the connection between the NAFDI and the SHL, quantifying their relationship, we need to introduce a daily NAFDI index along with another daily index that accounts for variations in the position of the SHL. In the new section 3.3 we also work with the daily NAFDI index in order to know its relationship with the daily variability of mid-latitude Rossby waves. Anyway, the terms "monthly" and "daily" are always explicitly stated to avoid any confusion.

MC3. All the results are presented without significance tests.

We agree with the Referee. This point had been raised by Referee 1 too, and we provide here the same arguments (already presented in our reply to Referee 1) and detail of changes are going to be implemented.

Concerning the statistical results of the NAFDI-SHL and Rossby Waves-NAFDI daily relationships addressed in Sections 3.3. and 3.4 for the summers of the period 1980-2013, we assumed that the fairly good Pearson correlations obtained using more than 3,000 pair-data implied a high statistical significance to "the naked eye". Given the great length of the paper (addressed by

the three Referees), we focused efforts in describing the physical mechanisms behind these relationships and document the statistical results. However, we agree with the Referee. A complete assessment of statistical significance is presented here for the main results of the paper. Please, indicate if you require specific additional minor results.

**a)** Concerning the correlation plots, the following sentence will be added in the manuscript (Section 3.1):

 "The correlation plots shown in this paper are computed using monthly means (only one month per year, e.g., August) of the period 1980-2013 (i.e., 34 years), except for Fig. S5 (2 monthly means per year). Therefore, the critical value for having a significant Pearson´s correlation coefficient ($R$) with a 95% confidence level is 0.34 (i.e., the correlation is significant if $|R|>0.34$)."

Correlation plots are shown in Figs. 1, 11, S5, S6 and S11. The following sentence will be included in the caption of all these plots: "Correlations greater (in absolute value) than 0.34 are significant with a 95% confidence level", except in Fig. S5, for which the critical value is 0.24. All the correlation plots shown have large regions where the correlation is well above this threshold.

**b)** Concerning the correlations between the two NAFDI index versions and the monthly total dust concentration measured at Izaña Observatory for August months (presented in Section 3.1), the number of elements in the time series is 26, and therefore, the critical value for having a significant Pearson´s correlation coefficient ($R$) with a 95% confidence level is 0.39 (i.e., the correlation is significant if $|R|>0.39$).

The correlation of the dust at Izaña Observatory with the old version of the NAFDI index is 0.67 ($r_1$; this value is significant with a 99.98% confidence level), whereas with the new version of the NAFDI index is 0.72 ($r_2$; this value is significant with a 99.997% confidence level). There is a confidence level of 63% about the fact that $r_2$ is significantly larger than $r_1$ .Some details about this confidence level computation will be presented in the Supplement, including the corresponding references; i.e., Fisher transformation, computation of the difference... However, the decision of providing an improved version of the NAFDI index is not based on the improvement of this correlation.

These three confidence levels will be provided in the main text of the manuscript.

 **c)** Concerning the correlations between the daily index time series (3,060 values per time series):

Assuming that there is no time-lag autocorrelation in any of the time series, the critical value for having a significant Pearson´s correlation coefficient ($R$) with a 95% confidence level is 0.036 (i.e., the correlation is significant if $|R|>0.036$).

However, indeed, there is time-lag autocorrelation in the time series. We have used the method exposed below to establish an upper bound for the critical value of the Pearson's correlation coefficient. In the four time series (NAFDI, SHLWEDI, O500 and ZWA300) the

autocorrelation decreases when increasing the time-lag. The maximum time-lag (MTL) in which there is still a significant autocorrelation (larger than 0.036) is: 22 days for the NAFDI time series, 23 days for the SHLWEDI time series, 7 days for the O500 time series, and 15 days for the ZWA300 time series. Then, to establish an upper bound to the critical value, we consider a lower bound of the number of independent values (LBNIV) in the time series, computed as the ratio between 3060 and MTL. Note that this is a very conservative estimation of the lower bound. For correlations including the SHLWEDI, LBNIV is 133, and the upper bound to the 95%-confidence-level critical values is 0.17. For correlations including NAFDI but not including SHLWEDI, LBNIV is 139, and the upper bound to the 95%-confidence-level critical values is 0.166. However, for correlations not including the former time series but ZW300, LBNIV is 204, and the upper bound to the 95%-confidence-level critical values is 0.137. These explanations will be included in the Supplementary Material.

The correlation ($r_2$) between the 1-day time-lagged SHLWEDI and the daily NAFDI is 0.770 (this value is significant with a 99.999% confidence level) whereas the non-lagged correlation ($r_1$) is 0.688 (this value is significant with a 99.999% confidence level). There is a confidence level of 92.2% about the fact that $r_2$ is significantly larger than $r_1$. Some details about this confidence level computation will be presented in the Supplement, including the corresponding references; i.e., Fisher transformation, computation of the difference…

These three confidence levels will be provided in the main text of the manuscript.

The paragraph starting in line 26 of page 26 will be rewritten as follows (the new text is highlighted here using bold face letter):

"*Table 4 shows the Pearson correlation coefficient between the daily ZWA300, O500 and NAFDI (also for some time lags as well as 5-day running means -5drm-). The results led to the following conclusions: 1) the correlation of both ZWA300 and O500 with NAFDI is significant **(with a confidence level larger than 99.999%)**; 2) the correlation between ZWA300 and O500 is low but not negligible **(with a confidence level larger than 86.5%, and 99.7% in case a 5drm is previously applied);** these two facts together indicate that ZWA300 and O500 are quasi-independent indexes that take into account different aspects of the Rossby wave in agreement with our previous discussion); 3) it seems that ZWA300 drives almost one day in advance the value of NAFDI (**the correlation with NAFDI lagged 1 day is larger than the correlation without any time lag, with a confidence level larger than 58.7%; the correlation with NAFDI lagged 1 day is larger than the correlation with NAFDI lagged -1 day, with a confidence level larger than 72.9%**), whereas O500 might be ahead respect to NAFDI less than 12 hours (**the correlation with NAFDI lagged 1 day is significantly larger than the correlation with NAFDI lagged -1 day, with a confidence level larger than 79.1%),** which agrees what is shown in Figures 5 and 7 of Chauvin et al. (2010): a Rossby wave-packet comes from the Northwest Atlantic and approaches Northern Africa days before a maximum in the SHL displacement is achieved, reaching the centre of the wave-packet Northern Africa when that maximum is achieved ; 4) when applying 5-day running means to the time series before computing the correlation coefficients, they increase significantly (because of the removal of part of the "noise" due to synoptic signal). When performing a multilinear least-square regression of daily*

*NAFDI as function of ZWA300 and O500, a linear correlation of 0.533 is obtained (0.656 for 5-day running means; **in both cases, the correlation is significant with a confidence level larger than 99.999%**). Supplement S13 provides more details about these regressions.*"

MC-4. The definition of the NAFTI is justified with the August correlation. Why June/July and September are not taken into account?

The reason is that the concept of NAFDI was introduced and used by Rodríguez et al. (2105) for August months. This is one of the central months of summer. So, in order to compare with previous results, we justify the slight change in the NAFDI index definition using only August months. In the new manuscript version, this change of the NAFDI index is presented not in the main paper but in the Supplement. We guess that the solid arguments we present are valid for the rest of the months of the summer period. The analysis for each month of the summer period would require an unnecessary (an unwanted) extension of the paper.

MC-5. September month is not provided in Fig. 2 to Fig. 9. But the authors discussed the link with the SHL that is defined for the entire rainy season over the Sahel from 20 June to 17 September. Why the NAFTI is studied only from June to August ? is there a scientific reason ? I agree that is this is the common definition of summer, but in a science point of view this is not necessary robust due to the seasonal cycle of the West African Monsoon.

We appreciate this suggestion and agree with Referee#3. The only reason for analyzing the period June-August is that this is the considered standard summer period, used in most papers. However, we agree that September is also a "summer month" to be account for in the study region. So we have included September in all our monthly analysis. Please, notice that September had been already considered on daily analysis.

The Referee will see in the results how the patterns found in AOD and meteorology for the two NAFDI phases in September are quite similar to those observed in July and August confirming the results found.

This is an example plot: AOD anomalies for negative and positive NAFDI phases.

[Figure]

MC-6. There is a constant back and forth between ERAI and NCEP reanalysis. I would suggest to use only one set of data to be consistent. These two reanalysis are relatively closed but difference are still present.

As we had already replied to the same comment posted by Referee#1, at synoptic scale, the scale at which we are working in this study, both ERA-Interim (ECMWF) and NCEP/NCAR reanalysis provide the same patterns. We decided to use ERA-Interim in some plots of limited geographical domain (new sections 3.1. and 3.2) because of its higher spatial resolution compared with NCEP/NCAR. On the contrary we found more practical to use NCEP/NCAR reanalysis for correlation and regression plots (with NAFDI) in larger geographical domains.

On the other hand, the use of two independent reanalysis has been carried out in other studies. For example, Knippertz and Todd (2010) used simultaneously ERA-Interim and NCEP/NCAR reanalysis. Chauvin et al. (2010) reported that no fundamental differences were found in their analysis of the intra-seasonal variability of the SHL and its link with mid-latitudes, when considering the 1948-2007 period of the NCEP-NCAR Reanalysis or the 1979-2001 period of ERA-40, even using different time periods.

Anyway, and in order that readers can verify that both reanalysis show the same large-scale patterns, the new Figures 4, 5, 6, 7, 8, 9 of the manuscript  plotted using ERA-Interim (which now also include September months) have been also plotted using NCEP/NCAR reanalysis and shown in the Supplement.

MC-7. The role of the African Easterly Waves, one of the most important component of the west african monsoon that can modulate the wind and temperature fields at 700hPa is not mentioned.

The West African Monsoon (WAM) is out of the scope of our study, and therefore it is very briefly mentioned in our manuscript. African Easterly Waves (AEWs) do play an important role modulating the precipitation in the WAM in their time scale (3-5 days; e.g., Holton, 1992). Moreover, due to their relatively short period (3-5 days), their imprint is not seen in monthly mean meteorological plots. Anyway, AEWs are not expected to have any significant impact on the SHL position as argued below.

The African Easterly Jet (AEJ) is the thermal wind associated to the strong temperature gradient between the equator and the SHL over West Africa. AEJ has its maximum wind speed near 650 hPa. AEWs form and propagate in the AEJ, obtaining their energy by barotropic and baroclinic conversions of energy from the AEJ (Holton, 1992). Therefore, the position of the SHL determines where the AEWs are formed. The SHL is the northern flank of the background state that determines the AEJ, being the AEWs perturbations to such background state. Then, the AEWs cannot produce a significant impact into the SHL, because for that it would be necessary the AEWs become highly no linear and a subsequent strong no linear interaction between these waves and the background state.

Lavaysse et al. (2010) showed empirically that AEWs have no significant impact on the SHL (see their Figure 12).

Since the manuscript contains already a lot of relevant material and it is rather long, we think it is better not including any mention to the AEWs. However, if the referee insisted again that the AEWs must be (extensively) mentioned in the manuscript, we would include the above arguments at the end of the new section 3.3.

References:

Holton, J. R.: An introduction to dynamic meteorology, Third edition, International Geophysics Series, vol. 48, Academic Press, San Diego, California, 1992.

Lavaysse, C., Flamant, C., Janicot, S., and Knippertz, P: Links between African easterly waves, mid-latitude circulation and the intra seasonal pulsations of the West African Heat Low, Quart. J. Roy. Meteor. Soc., 136, 141 – 158, DOI: 10.1002/qj.555, 2010b.

MC-8. The use of the geopotential at 1000 hPa should be used with a lot of caution due to the topography of the region. The value is provide by using an linear interpolation technique that can influence the results.

We only used the 1000 hPa level for temperature anomalies analysis. We have replaced 1000 hPa by 925 hPa, the lowest level considered no very affected by the topography, in the new manuscript version. As the Referee can see, the results are quite similar in both levels.